# Old and New Drugs for Chronic Lymphocytic Leukemia: Lights and Shadows of Real-World Evidence

**DOI:** 10.3390/jcm11082076

**Published:** 2022-04-07

**Authors:** Monia Marchetti, Candida Vitale, Gian Matteo Rigolin, Alessandra Vasile, Andrea Visentin, Lydia Scarfò, Marta Coscia, Antonio Cuneo

**Affiliations:** 1Haematology and Transplant Unit, Azienda Ospedaliera SS Antonio e Biagio e Cesare Arrigo, 15121 Alessandria, Italy; 2Hematology Unit, Città della Salute e della Scienza, University of Turin, 10126 Turin, Italy; candida.vitale@unito.it (C.V.); marta.coscia@unito.it (M.C.); 3Haematology Unit, Azienda Ospedaliera Universitaria di Ferrara, 44121 Ferrara, Italy; rglgmt@unife.it (G.M.R.); cut@unife.it (A.C.); 4Haematology and Rheumatology Section, Department of Medical Sciences, University of Ferrara, 44121 Ferrara, Italy; 5Department of Public Health, University of Eastern Pedemont, 28100 Novara, Italy; 20034878@studenti.uniupo.it; 6Hematology and Clinical Immunology Unit, Department of Medicine, University of Padua, 35128 Padua, Italy; andrea.visentin@aopd.veneto.it; 7Division of Experimental Oncology, Department of Onco-Hematology, IRCCS San Raffaele Hospital, 20132 Milan, Italy; scarfo.lydia@hsr.it

**Keywords:** chronic lymphocytic leukemia, ibrutinib, idelalisib, venetoclax, acalabrutinib, real-world evidence

## Abstract

Several novel treatments for chronic lymphocytic leukemia (CLL) have been recently approved based on the results of randomized clinical trials. However, real-world evidence (RWE) is also requested before and after drug authorization in order to confirm safety and to provide data for health technology assessments. We conducted a scoping review of the available RWE for targeted treatments of CLL, namely ibrutinib, acalabrutinib, idelalisib, and venetoclax, as well as for chemoimmunotherapy (CIT). In particular, we searched studies published since 1 January 2010 and reported outcomes of the above treatments based on health databases, registries, or phase IV studies, including named-patient programs. We included both full papers and abstracts of studies presented at major meetings. Overall, 110 studies were selected and analyzed: 28,880 patients were treated with ibrutinib, 1424 with idelalisib, 751 with venetoclax, 496 with acalabrutinib, and 14,896 with CIT. Reported discontinuation rates were higher than in clinical trials, while effectiveness could not be indirectly compared with clinical trials since a detailed case mix, including cytogenetic risk factors, was partially available and propensity scores rarely applied. RWE on CLL can help to set realistic outcomes with novel treatments, however, real-world studies should be fostered, and available data shared.

## 1. Introduction

Chronic lymphocytic leukemia (CLL) is an indolent lymphoproliferative neoplasm harbored by 5.6 in 10,000 inhabitants (https://seer.cancer.gov/statfacts/html/clyl.html accessed on 30 December 2021). Clinical practice guidelines formerly recommended frontline chemoimmunotherapy (CIT) for all the patients [1,2,3], but several novel treatments, along with several combinations, have been progressively approved in the last 5 years by FDA and EMA: ibrutinib, idelalisib, venetoclax, obinutuzumab, acalabrutinib. The current therapeutic alternatives are, therefore, more diverse than in the past and several concerns apply on patient selection to a personalized treatment sequence. Moreover, patient selection bias due to severe restrictions of patient eligibility to clinical trials is a major hurdle for evidence-based medicine and health technology assessment of novel drugs [4]. Real-world evidence (RWE) gathered either retrospectively or prospectively from electronic health records, medical claims, databases, registries, or patient-generated data can complement the information reported by experimental studies. The pivotal role of RWE is witnessed by the 21st Century Cures Act requiring the FDA to expand the role of RWE and by the European Medicines Agency (EMA) draft guideline on real-world studies (RWS) developed either in the pre- or post-marketing authorization phase of drugs and devices (Table 1). (EMA draft guideline on real world studies 24 September 2020). 

The present scoping review aims at scrutinizing RWE for novel targeted treatments of CLL [5,6], namely ibrutinib, acalabrutinib, idelalisib, and venetoclax, as well as CIT. In particular, we aimed at checking the quality of available registries and named-patient program (NPP)-based studies, patient selection bias (such as age or comorbidities), the quality of available clinical and non-clinical data, and the length of follow-up. 

## 2. Methods

We searched EMBASE, the largest bibliographic database of medical literature, by applying the following mast query: ‘chronic lymphocytic leukemia’ AND (RWD OR ‘real life’ OR ‘real world’ OR EHR OR registry OR register OR registries OR ‘phase IV’ OR post-marketing OR population-based) AND (bendamustine OR fludarabine OR chlorambucil OR venetoclax OR ibrutinib OR idelalisib OR rituximab OR ofatumumab OR obinutuzumab OR acalabrutinib OR rituximab) AND (2010:py OR 2011:py OR 2012:py OR 2013:py OR 2014:py OR 2015:py OR 2016:py OR 2017:py OR 2018:py OR 2019:py OR 2020:py OR 2021:py) AND ‘article’/it. The search was limited to English-language publications reported from 1 January 2010 to 31 December 2020.

Two independent reviewers excluded the inappropriate records and retrieved the data from the selected records. Unspecific reports were deleted. Studies were also excluded if targeted at etiology (familiar lymphoproliferative disorders and exposures), survival, and prognosis (molecular biomarkers and undesirable events). 

The following information were retrieved from selected records: 

Geographic limits

(a)international,(b)national extensive,(c)national multi-site,(d)regional multi-site,(e)single-institution

Data source 

(a)health databases,(b)existing registry,(c)newly developed registry,(d)retrospective chart review,(e)phase IV study

Population selection

(a)geographic,(b)treatment,(c)sample availability,(d)frailty,(e)number of lines

Population size 

(a)<50,(b)≥50

Outcomes 

(a)practice patterns,(b)survival,(c)prognostic yield of biomarkers,(d)prognostic yield of a score,(e)patient-related outcomes (PRO),(f)health care resource consumption and costs,(g)adverse events and discontinuation

## 3. Results

Overall, 433 records were retrieved (Figure 1) and showed that the overall trend of publications addressing RWE of CLL rapidly had increased in the last 10 years (Figure 2). The full reference list is reported in Appendix A. Overall, 102 studies were selected from the list and 8 studies were added from the authors, therefore, the search query proved to be quite sensitive. Five studies were retrieved for more than one treatment. The full list of selected RWS is included in Appendix A. Major study characteristics are reported in Table 1. Most of the studies were from single institutions and size was often lower than 500 patients. Patients were representative of the real-life CLL population according to age. However, collected clinical data were unfortunately sparse in most of the published studies, for example, comorbidities were rarely systematically recorded and response rates were not reported by most of the studies. Moreover, follow-ups were short for most of the studies involving new oral inhibitors.

Only 1 phase IV study was reported by Moreno, C. et al. [7]; the study reported the outcomes of 103 patients treated with ofatumumab monotherapy after a median of four prior lines. The study reported a 5-month progression-free survival (PFS) and 11-month overall survival (OS), thus, confirming the data of the published randomized trial comparing ibrutinib versus ofatumumab.

### 3.1. Ibrutinib

Ibrutinib is an orally administered irreversible inhibitor of bruton tyrosine kinase (BTK), representing the first-in-class of a new family of targeted drugs. Ibrutinib covalently binds to cysteine-481 within the active site of BTK, blocking the signal transduction from the B-cell receptor (BCR) and, thus, impairing CLL cells survival and proliferation (reviewed in [8]). Ibrutinib has changed the therapeutic approach for patients with CLL both in first and subsequent lines of therapy, due to impressive response and survival rates, and acceptable tolerability observed in large, randomized clinical trials [9,10,11,12,13].

Many RWS addressed ibrutinib. Overall, 58 nonduplicated reports enrolled more than 50 CLL patients treated with the drug, mostly in the last 10 years. Enrolled populations were potentially representative of the clinical practice: median age was 69 years and median time from diagnosis was 56 months. Unfortunately, only a few studies were powered enough to detect predictors of safety or effectiveness endpoints in a multivariate analysis, namely large registries in the U.S. or healthcare databases. Moreover, both naive and relapsed/refractory patients were usually included. Many pieces of information were missing from many studies: median doses or median treatment durations, response rates, survival rates, and *TP53* mutational status. Several other flaws could be observed in the retrieved studies: limited outcomes were analyzed (i.e., safety or hospitalization rate) and there were no studies that applied propensity-adjusted analyses. Despite the above limitations, the studies provided high discontinuation rates in the short time horizon analyzed. 

### 3.2. Acalabrutinib

Acalabrutinib is a next-generation irreversible BTK inhibitor that was developed to reduce ibrutinib-mediated adverse effects, being more selective and lacking the inhibition towards other kinases (reviewed in [14]). Acalabrutinib has entered clinical practice based on data demonstrating its high efficacy and enhanced tolerability, in both the frontline and relapsed/refractory setting [15].

We retrieved four RWS reporting 496 patients with CLL treated with acalabrutinib; only one RWS was fully reported. A variable preportion of the patients (27–100%) were pretreated with and usually intolerant to ibrutinib. The discontinuation rate reported by two studies ranged from 19% in the first 6 months from treatment initiation to 30% after a median of 19 months. Cardiovascular events occurred in 6% of the largest reported cohort, but led to treatment discontinuation only in half of the cases and corresponded to a rate of 21/1000PY, which was lower than that reported in ibrutinib-treated patients [16].

The overall response rate (ORR) was consistently above 60% in two studies and complete response (CR) was lower than 10%. Survival was documented in two cohorts (median age 64 years); 75% of the patients survived free of progression at three years and 75% were alive after five years, despite a high comorbidity burden, namely mean Charlson comorbidity score 1.4 and 67% of the patients had a prior cardiac history. Patients experiencing a major cardiovascular event reported lower survival rates, namely 50% at 5-year follow-up.

Long-term outcomes such as secondary neoplasms were also investigated in patients exposed to acalabrutinib, however, the increased hazard ratio of 2.2 reported in CLL patients treated with either ibrutinib or acalabrutinib was possibly related to the disease itself, rather than to the treatment. No difference in risk of secondary neoplasms was reported in a multivariate analysis between ibrutinib and acalabrutinib.

A recent retrospective analysis of a large CLL cohort from a single U.S. center specifically investigated the bleeding outcomes [17], however, 85% of the analyzed patients had been enrolled into clinical trials, 18% had prior bleed history, and 51% were on concomitant antiplatelet or anticoagulant medications. Overall, 835 of the patients experienced at least one bleeding event while on acalabrutinib; 98 out of 289 individuals experienced a clinically relevant or major event, but only 6% of the patients had a major bleed and 3% were CTCAE grade 3–5 (2 CNS fatal hemorrhages). Definitive discontinuation of acalabrutinib was decided for 6 patients with clinically relevant/major bleeds, while it was only temporarily held in 44 individuals and concomitant drugs were discontinued in 24 cases. Surgery- or invasive procedure-related bleedings were reported in 28 out of 1263 cases. Concomitant medications and a prior bleeding history were major predictors of bleeding events.

### 3.3. Venetoclax

Venetoclax is an oral BH3-mimetic drug designed to inhibit the function of the Bcl-2 protein, thus, inducing apoptosis in tumor cells (reviewed in [8]). Venetoclax, alone or in combination with an anti-CD20 monoclonal antibody, has demonstrated efficacy for the treatment of patients with treatment-naïve or relapsed CLL, eventually allowing a fixed-duration treatment [18,19,20].

Seven studies reported RWE on venetoclax in CLL patients; one study reported the French national early-access program and the other studies were retrospective national (*n* = 1) or multicenter (*n* = 5) studies. Overall, 751 patients were enrolled into the above studies, most of which were not reported as full papers. Enrolled patients had received a median of from three to four treatment lines, 47% harbored *TP53* mutation, and many showed high-risk features, such as complex karyotype (27–61%) or unmutated IGVH status (81–87%). Grade 3–4 adverse events ranged from 23% to 39% and were mainly due to hematologic toxicity. Discontinuation rate was reported only by two studies and was quite low (4–11%); median treatment duration ranged from 12 to 18 months in two other studies. Response rates were quite high: median ORR was 74% and median CR was 25%. Richter transformation occurred in 4–5% of the patients but was reported only by two studies and the median duration of follow-up was shorter than 20 months in the four studies reporting this piece of information. Median PFS and OS were not reached in any study. Multivariate analyses were performed by five studies: survival was predicted by response to therapy, *TP53* mutation (in two out of three studies), *BCR*-inhibitor discontinuation, multiple lines of target therapies, complex karyotype, performance status, and *IGVH* mutational status. 

### 3.4. Chemoimmunotherapy

Traditionally, CIT was the standard approach in both the frontline and the relapsed/refractory setting of CLL. Common CIT regimens include fludarabine/cyclophosphamide/rituximab (FCR), bendamustine/rituximab (BR), and chlorambucil/rituximab. Older patients or those with comorbidities were recommended to not receive FCR due to the high risk of neutropenic fever and infections, despite very high rates of response in most of the patients [1,2]. More recently, obinutuzumab combination with chlorambucil has progressively replaced chlorambucil/rituximab for the longer median PFS and OS despite similar toxicity, as reported by the CLL14 trial [21,22].

We retrieved 428 studies assessing CIT in real life by health registries (*n* = 13), electronic record databases (*n* = 2), or retrospective data collections (*n* = 11). Most of the studies involved multiple centers (*n* = 15) and followed a median of 277 patients (IQR 174–817) for a median of 37 months (Table 2 and Table 3). The median patient age was 70 years (IQR 64–71) and most of the studies (*n* = 18) selectively enrolled naive individuals, but clinical information was often incomplete (Table 4). In particular, Rai stage and *TP53* status were reported by 64% and 75% of the studies, respectively, and comorbidity data were missing in 61% of the studies. Response rates were reported by only 13 studies: median ORR rate was 61% and complete response (CR) rate 6%. Discontinuation rate was reported by only five studies: median rate was 20%. Patient survival was described by 75% of the studies (Table 3): median PFS and OS among the studies was 42 and 74 months, respectively.

### 3.5. Idelalisib

Idelalisib is an orally administered, selective, reversible inhibitor of the δ isoform of the phosphatidylinositol- 3-kinase (PI3Kδ). The inhibition of *PI3K* downstream pathways (i.e., *Akt*/*mTOR*) hampers cellular growth, proliferation, and survival (reviewed in [23]). From the clinical standpoint, the efficacy of the combination of idelalisib and rituximab has been demonstrated in the setting of relapsed CLL [8].

Overall, 16 studies reported one or more outcomes of real-world cohorts treated with idelalisib; most of the reported patients were relapsed/refractory and more than one third harbored *TP53* disruption. 

The largest cohort was reported by Bird et al. [24]; 294 Medicare patients were compared with 89 patients enrolled into clinical trials. Detailed comorbidity profiles of the patients were provided showing that in the two cohorts, 71% versus 30% of the patients reported cardiac comorbidities, respectively. Similarly, 36% versus 7% showed a Charlson comorbidity score of 5 or higher. Median treatment duration in the Medicare cohort was much shorter than in the trial cohort, namely 173 versus 473 days and on-treatment mortality was 9.9% versus 4.5%. Serious infections were not different in the two cohorts, however, fatal infections occurred at a rate of 18.4/100PY in the Medicare population versus 9.8/100PY in the trial population.

RETRO-idel was the highest quality retrospective study selected; it enrolled 110 patients treated with idelalisib in UK or Ireland, and was fully published in 2021 [25]. The study reported high discontinuation rates both in naiive and relapsed/refractory patients, namely 64% and 44%, respectively, but high ORRs (88%). Overall, 46% of the registered deaths were attributed to progressive disease and OS was 56% after 3 years. Median time-to-next treatment after stopping idelalisib was 29 months.

Two studies were specifically designed to patients reporting autoimmune cytopenias before starting idelalisib or receiving such a therapy as a bridge to stem cell transplantation [26,27]. Only one study reported secondary neoplasms in 12.9% of the patients after a median of 21 months from start of idelalisib treatment; the rates were not statistically different to those reported in ibrutinib-treated patients [28].

## 4. Discussion

CLL is the most frequent leukemia and a large set of modern therapeutic options are available ranging from CIT to oral targeted drugs. These therapeutic options have all reported high rates of responses but also a relevant rate of toxicity in the published clinical trials, which accurately selected candidate patients. We, therefore, aimed at reviewing the available RWE on such therapies in order to test the quality of RWE and to retrieve real-world information for safety and effectiveness. 

EMA specifically fostered the development of patient registries, namely data collection systems on an unselected group of people defined by a particular disease or condition, serving a predetermined scientific, clinical. and/or public health purpose. RWE is particularly relevant for validating safety, including cardiovascular events, and especially for reporting rare events, such as SPM, rare infections, or unexpected events. RWE is also necessary for completing drug effectiveness profiles, including rare events, such as Richter transformation and concurrent disorders. Furthermore, treatment outcomes according to heterogeneous adoption of supportive care are fundamental in order to forecast the overall drug safety in the real world and provide management recommendations [29].

The present manuscript systematically reviewed 117 RWS published from 2010 to 2021 and reported 46,447 CLL patients treated with CIT, idelalisib, ibrutinib, venetoclax or acalabrutinib. Unfortunately, 77 studies had been reported only at meetings and only limited data were available. Most of the studies were multicenter retrospective analyses and most of them targeted only a subset of outcomes. A complete clinical dataset, including age, stage and duration of the disease, comorbidity, and biologic risk status was provided only by a few studies, in particular, stage and disease duration were missing in most of the studies, while *TP53* status was available in 57% of the studies. Only 26% of the RWS reported treatment discontinuation rates and 40% registered response rates; the median response rates ranged from 61% for acalabrutinib to 79% for idelalisib, and CR rate from 6% for acalabrutinib to 30% for CIT. Survival was also poorly described, since median OS or PFS were usually not reached in the follow-up period, which was usually shorter than 2 years in RWS assessing oral target drugs. Of notice, some research networks devoted to CLL published further high-quality RWS both before [30,31] and after the cutoff date of our review [32,33,34,35,36], thus, demonstrating the effort of the scientific community in ameliorating the RWS quality.

The present systematic review aims at fostering further efforts of the scientific community towards RWE, which may become a very useful tool for both researchers and third-party payers. Institutional databases are major prerequisites for RWE, however, further efforts should be aimed at registering response rates, time-to-next treatment, and comorbidities [37]. Finally, large datasets would allow fine analyses including artificial intelligence algorithms and data mining.

## Figures and Tables

**Figure 1 jcm-11-02076-f001:**
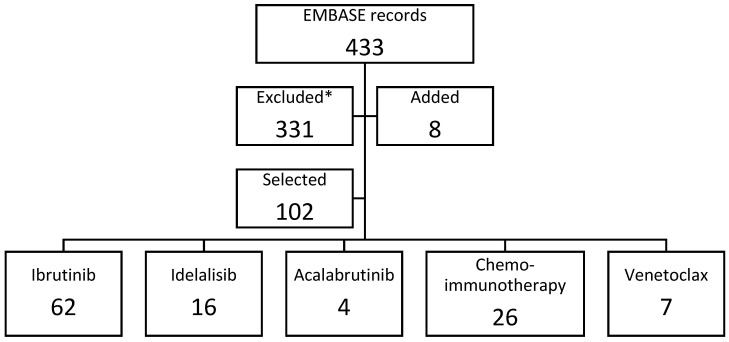
PRISMA diagram.* Reasons for exclusion: less than 20 patients (acalabrutinib), less than 50 patients (other series), mixed treatments, missing outcomes, review.

**Figure 2 jcm-11-02076-f002:**
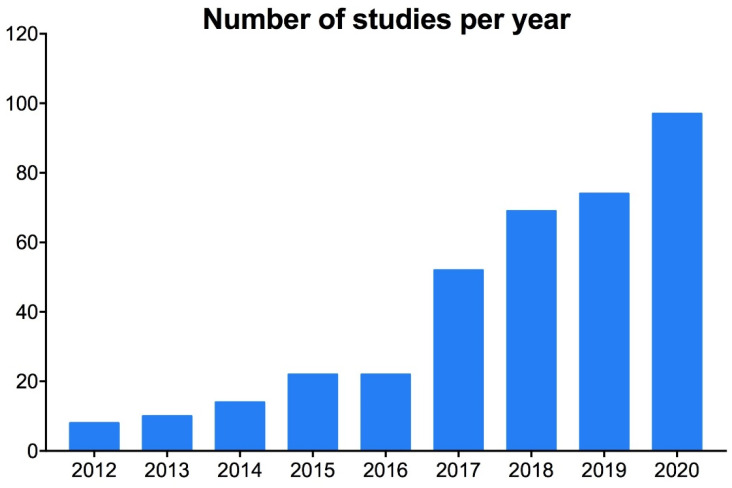
Publication trends of RWS. Distribution of the 433 retrieved RWS according to the year of publication.

**Table 1 jcm-11-02076-t001:** Aims of real-world studies (RWS).

Purposes of RWS before Drug Authorization
To describe the characteristics of the target population
To record the incidence of disease outcomes in the clinical practice
To identify the determinants of disease outcomes in the clinical practice
To provide information on standards of care
**Purposes of RWS after Drug Authorization**
To confirm safety and effectiveness in the target population (i.e., phase IV studies)
To confirm safety in subpopulations (i.e., comorbid patients)
To survey modified patterns of care and health-care resource consumption

**Table 2 jcm-11-02076-t002:** Real-world studies.

	All	Ibrutinib	Idelalisib	Venetoclax	CIT	Acalabrutinib
Study number	117	62	16	7	28 ^	4
Registry	34 (29%)	18	3	0	13	-
NPP/EAP	13 (11%)	10	2	1	-	-
Electronic record database	33 (28%)	28	2	0	2	1
Retrospective data collection	70 (60%)	45	5	6	11	3
Multicenter	52 (44%)	21	10	5	15	1
Europe	40 (45%)	25	11	4	na	-
US/international	41 (46%)	29	5	3	na	4
No explicit patient selection	52 (63%)	37	12	na	na	3
Selectively naiive patients	27 (23%)	8	0	0	18	1
Full papers	40 (34%)	15	5	2	17	1

^ 28 populations from 25 published reports. Legend: NPP, named patient program; EAP, expanded access program; CIT, chemo-immunotherapy; na = not available.

**Table 3 jcm-11-02076-t003:** Information reported by real-world studies.

Studies Reporting:	All	Ibrutinib	Acalabrutinib	Venetoclax	Idelalisib	CIT
Number of treated patients	113 (96%)	58 (93%)	4 (100%)	7 (100%)	16 (100%)	28 (100%)
Patients’ age	90 (77%)	48 (77%)	4 (100%)	5 (71%)	10 (62%)	25 (89%)
Number of treatment lines	61 (52%)	21 (34%)	2 (50%)	5 (71%)	9 (56%)	24 (86%)
Rai/Binet stage	44 (37%)	17 (27%)	1 (25%)	2 (28%)	6 (37%)	18 (64%)
Median time from diagnosis	11 (9%)	8 (13%)	1 (25%)	0	2 (12%)	na
Comorbidity	23 (20%)	7 (11%)	2 (50%)	1 (14%)	2 (12%)	11 (39%)
*TP53* status	67 (57%)	32 (52%)	2 (50%)	5 (71%)	7 (44%)	21 (75%)
Other high-risk molecular or cytogenetic features	61 (52%)	33 (53%)	2 (50%	5 (71%)	2 (12%)	19 (67%)
Median follow-up	78 (66%)	38 (61%)	2 (50%)	4 (57%)	8 (50%)	26 (93%)
Discontinuation rate	31 (26%)	13 (21%)	2 (50%)	2 (28%)	9 (56%)	5 (18%)
Response rate	47 (40%)	20 (32%)	2 (50%)	6 (86%)	6 (37%)	13 (46%)
Richter transformation	10 (8%)	6 (10%)	0	2 (28%)	2 (12%)	na
PFS	52 (44%)	22 (35%)	1 (25%)	4 (57%)	4 (25%)	21 (75%)
OS	61 (52%)	25 (40%)	1 (25%)	5 (71%)	8 (50%)	22 (78%)
TFS or TTNT	16 (14%)	2 (3%)	0	0	2 (12%)	12 (43%)
SPM	12 (10%)	2 (3%)	1 (25%)	0	1 (6%)	8 (28%)
Specific adverse events ^	12 (10%)	4 (6%)	1 (25%)	3 (57%)	4 (25%)	0

^ bleedings and atrial fibrillation for Ibrutinib, cholitis for idelalisib. Legend: CIT, chemoimmunotherapy; SPM, secondary primary malignancies; OS, overall survival; PSF, progression-free survival; TSF, threatment-free survival; TTNT, time-to-next treatment; na = not available.

**Table 4 jcm-11-02076-t004:** Patient characteristics.

	Ibrutinib	Idelalisib	Venetoclax	CIT	Acalabrutinib
Patient number: Mean, median, inter-quartile range	486 179 89–554	89 74 54–104	107 76 67–149	532 277 174–817	149 136 33–290
Age: Median years, inter-quartile range	69 65–70	72 67–74	68 67–69	70 68–75	68 64–71
Number of prior treatment lines: Median, inter-quartile range	2 0–3	3 1–4	3.5 3–4	0 0–1	4 (1 study)
Follow-up (mo): Median, inter-quartile range	16 9–21	16 7–19	13 9–17	37 25–57	12 5–19
Discontinuation rate: Range	23–41%	63–100%	7–27%	2–30%	19–30%
Overall Response Rate: Median, inter-quartile range	77% 73–84%	79% 65–86%	74% 72–75%	83% 76–91%	61% 60–62%
Complete Response Rate: Median, Inter-quartile range	17% 11–18%	14% 1 study	25% 23–28%	30% 19–55%	6% 3–9%
PFS: Median, range,	38 na	na 10–36	na na	42 28–51	na na
OS	na *	>36	na	74	na

* median OS was usually not reached in the median follow-up reported. Median reported values were usually >48 months for first-line treatment. na = not available.

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
