# Peer review of "Old and New Drugs for Chronic Lymphocytic Leukemia: Lights and Shadows of Real-World Evidence"

_jcm, 2022, doi:10.3390/jcm11082076_

Round 1

Reviewer 1 Report

The review manuscript by Monia, M. et al. submitted to Journal of Clinical Medicine entitled “Old and new drugs for chronic lymphocytic leukemia: lights and shadows of real world evidence” aims scrutinizing RWE the novel targeted treatments of chronic lymphocytic leukemia. In particular for checking the quality of available registries and named-patient program (NPP)-based studies, patient selection bias (such as age or comorbidities), the quality of available clinical and non-clinical data and the length of follow-up. In general the manuscript is well written, but I would like to suggest to the authors that a short paragraph be included informing the molecular targets of drugs and the role of these targets in the biology of CLL.

Author Response

The reviewers kindly suggested to complete the manuscript by adding " a short paragraph be included informing the molecular targets of drugs and the role of these targets in the biology of CLL."

We, therefore, proceded into completing each drug-specific result section with a brief introduction on the single therapeutic regimens (14 references added: in blish in the reference list).

Reviewer 2 Report

The review details on the RWE available for drugs used for treatment of CLL namely ibrutinib, acalabrutinib, idelalisib, venetoclax, and CIT. RWE is the need of the hour for assessing how drugs work post clinical trials, and in the clinic. Authors have done impressive work in searching, collecting, filtering and curating data. Authors have done exceptional job in presenting the data in succinct form and it will certainly attract scientific/clinical community working on CLL.

However, the authors could answers/fix the following comments

Comments

  1. Figure 1 needs to be presented well. I suggest authors remove the blue box which contains the numbers. Showing the number with an increased font is enough. In the same figure follow a uniform font, and use the capital letters.
  2. Figure 2 can be plotted in GraphPad for a better visual appearance.
  3. References selected for the EMBASE query can either be alphabetical or drug wise or chronological order.
  4. In multiple places there is an extra space after the period, please check and edit them.

Author Response

The reviewer kinldy sugegsted to revised Figure 1 and Figure 2: please find the figures revised accordingly. 

We also orderd Excel references alphabetically, according to the reviewer's sugegstion.

We also deleted 11 extra spaces.

Reviewer 3 Report

Article entitled „Old and new drugs for chronic lymphocytic leukemia: lights and shadows of real world evidence” is a summary of real world evidence about CLL treatment. In my opinion, the choice of topic is interesting and relevant. It should provide guidance on what reviewers should require from investigators describing data from clinical practice.  

I have small comments:

  1. Line 103 – Ofatumumab word missed?
  2. Ibrutynib - In my opinion, this part needs to be expanded.In other drug descriptions, information on ORR or side effects is given.As Ibrutinib is one of the most well-known drugs, I think it should receive more attention.
  3. Chemoimmunotherapy – line 177 - It seems to me that information about the treatment regimen we are talking about should be included.The differences in response rates and toxicity are significant between the O-Cl, FCR, BR regimens.
  4. Table 2 – Legend is missing, NPP, EAP, CIT

Author Response

The reviewer suggested to expand the section on ibrutinib and we thus added an introductory paragraph. We also followed the reviewr's suggestion to comment on chemoimmunotherapy and added an introductory paragraph, accordingly. 

Overall 14 new references were added (blish in final reference list). 

We added a legend to Table 2, which was missing.

We corrected line 103.